# Subcutaneous Implantable Cardioverter Defibrillators for the Prevention of Sudden Cardiac Death: Pediatric Single-Center Experience

**DOI:** 10.3390/ijerph191811661

**Published:** 2022-09-16

**Authors:** Piotr Wieniawski, Michał Buczyński, Marcin Grabowski, Joachim Winter, Bożena Werner

**Affiliations:** 1Department of Paediatric Cardiology and General Paediatrics, Medical University of Warsaw, 02-091 Warsaw, Poland; 2Department of Cardiac and General Paediatric Surgery, Medical University of Warsaw, 02-091 Warsaw, Poland; 31st Department of Cardiology, Medical University of Warsaw, 02-091 Warsaw, Poland; 4Division of Cardiac Surgery, University of Düsseldorf, 40225 Düsseldorf, Germany

**Keywords:** subcutaneous ICD, S-ICD, children, implantable cardioverter defibrillator, sudden cardiac death prevention, leadless ICD, SCD primary prevention

## Abstract

Background: The subcutaneous implantable cardioverter defibrillator (S-ICD) was developed as an alternative to the transvenous ICD, to prevent lead-related complications associated with the latter. The absence of intravascular or intracardiac components offers potential advantages to pediatric patients. Aims: The aim of the study is to present an overview of our experience with S-ICDs in the pediatric center that, currently, has performed the largest number of implantations in children in Poland. Methods: Retrospective analysis of data from medical history, qualification, implantation procedure, and S-ICD post-implantation observations in 11 pediatric patients were performed. Results: S-ICDs were implanted in 11 patients, 8 boys and 3 girls, aged 12–17 years. The S-ICD was implanted for primary prevention in seven patients: four with hypertrophic cardiomyopathy (HCM), two with dilated cardiomyopathy (DCM), and one with arrhythmogenic right ventricular cardiomyopathy (ARVC). It was implanted for secondary prevention in four patients: two with sudden cardiac arrest (SCA) in the course of idiopathic ventricular fibrillation (IVF), one with long QT syndrome (LQTS) after probable SCA, and one with Brugada syndrome after SCA. In all patients, the device was implanted intramuscularly. One patient did not have a defibrillation test performed due to the presence of an intracardiac thrombus. In one patient, during screening, it was decided to implant an electrode on the right side of the sternum. There were no early or late complications with any of the procedures. So far, no inadequate discharges have been observed. Conclusions: Our results prove the efficacy of the S-ICD treatment option along with technically simple surgery, which supports its further and more widespread application in children.

## 1. Introduction

Prevention of sudden cardiac death (SCD) is of major concern in cardiology. The development and spread of implantable cardioverter defibrillators (ICDs) have resulted in a breakthrough in treatment and prevention in patients at high risk of SCD. Currently, ICD implantations are considered routine procedures in cardiology departments in developed countries. To date, the majority of high-voltage devices are represented by transvenous leads. Less frequently, epicardial defibrillation patches are used, mainly in the pediatric population [1,2]. The most important advantage of the transvenous ICD (TV-ICD) is that it is a relatively safe, minimally invasive surgery, with a low complication rate. Nonetheless, even rarely observed lead-dependent problems may be life threatening, i.e., cardiac tamponade or infective endocarditis [3,4,5,6]. A totally subcutaneous ICD (S-ICD) was introduced as an aid for prevention of SCD. The leads of the S-ICD are implanted subcutaneously on the chest and a pulse generator is placed on the left lateral part of the chest subcutaneously or intramuscularly. During the procedure, the vascular system remains intact. The main disadvantage of S-ICD is its inability to provide permanent cardiac pacing [7,8]. The first S-ICD implantation in Poland was performed in October 2014 on a male adult who had a T-ICD removed and an artificial tricuspid valve implanted due to fulminant endocarditis [9]. S-ICDs are increasingly used also in the pediatric population.

In this retrospective study, we present a 4-year single-center experience with the use of S-ICDs. The aim of the study was to provide an overview of early experience with S-ICDs in the pediatric center.

## 2. Materials and Methods

Patients: 11 pediatrics patients, 8 boys and 3 girls, aged from 12 to 17 years were analyzed.

An S-ICD was implanted for primary prevention in 7 patients: 4 with hypertrophic cardiomyopathy (HCM), 2 with dilated cardiomyopathy (DCM), and 1 with arrhythmogenic right ventricular cardiomyopathy (ARVC). An S-ICD was implanted for secondary prevention in 4 patients: 2 with sudden cardiac arrest (SCA) in the course of idiopathic ventricular fibrillation (IVF), 1 with long QT syndrome (LQTS) after probable SCA, and 1 with Brugada syndrome after SCA. None of the patients had an absolute indication for permanent pacing.

Data from medical history, qualification, implantation procedure, and S-ICD post-implantation observations in 11 pediatric patients were analyzed. All patients were informed of the characteristics of the new system, indications, and potential complications. Informed consent was obtained from each patient. Prior to the procedure, patients were screened with a manual screening tool to assess the applicability of this technology. All patients were screened in accordance with currently applicable procedures and standards.

An electrophysiology expert participated in all the procedures. All procedures were performed under general anesthesia. Proctor Prof. Joachim Winter, a cardiac surgeon at the Medical University of Dusseldorf, participated in the first implantation procedure. The subcutaneous or intramuscular implantation technique and pocket placement developed by Professor Winter is considered to offer good aesthetic results, especially in young people, as well as to reduce local complications associated with the implant [10]. In all cases, the pocket for the generator was created between the serratus anterior and the latissimus dorsi muscles. The lead was positioned in the subcutaneous tissue of the chest, parallel to and 1–2 cm from the left sternal midline, then perpendicularly at the level of the 6th rib, until it reached the pocket of the device. The lead had an 8 cm shock coil, flanked by 2 sensing electrodes—the distal one positioned close to the manubriosternal joint and the proximal one adjacent to the xiphoid process based on preprocedural X-ray imaging. Immediately after implantation, the most suitable sensing vector (primary, secondary, or alternate) was chosen automatically by the S-ICD. The detection was set for one or two zones (a conditional shock zone and shock zone depending on the patient’s indication and condition). The duration of the procedure, defined as the total implantation time (time from patient in to patient out), and in-hospital adverse events related to the procedure were evaluated.

## 3. Results

The results are summarized in Table 1 and Table 2.

The S-ICD was implanted for primary prevention in seven patients:A 17-year-old male patient with hypertrophic cardiomyopathy with left ventricular outflow tract obstruction (HCM with LVOTO). Echocardiography revealed the following: significant degree of concentric myocardial hypertrophy, LVOTO with maximal systolic pressure gradient LV-Ao 108–112 mmHg, and mean pressure gradient 55 mmHg.A 17-year-old male patient with heart failure in the course of dilated cardiomyopathy, with gradually worsening left ventricular function and coexisting complex ventricular arrhythmia (VT episodes up to 170/min) and ventricular tachycardia (VT) with celiac disease, congenital IgA deficiency, and history of renal failure episode. The patient was qualified for heart transplant and LVAD (left ventricular assist device). A HeartMate 3 was implanted for left ventricular support as a bridge for heart transplantation. At the time of publication, the patient has not yet received a heart transplant. He is doing well, helps with lighter household chores, meets with friends, has a girlfriend, and describes his quality of life as satisfactory (Figure 1).A 17-year-old male patient with hypertrophic cardiomyopathy, obesity, insulin resistance, and arterial hypertension. Echocardiography revealed asymmetric LV hypertrophy and LV outflow tract obstruction with systolic pressure gradient LV-Ao up to 70 mmHg and complete obliteration of LV lumen during systole, hyperkinesis of both ventricles, mitral insufficiency, and left atrial enlargement. Non-sustained ventricular tachycardia (nsVT) was recorded in Holter ECG monitoring. During screening, correct sensing parameters were obtained for the electrode located on the right side of the sternum (Figure 2).

Despite obesity and the need to implant the electrode at a greater distance from the defibrillator, there were no procedure-related complications.

4.A 17-year-old male patient with dilated cardiomyopathy and severe heart failure, who was qualified for heart transplant. In echocardiography, significantly enlarged LV with impaired global myocardial contractility—LV EF 10–25%—and enlarged LA were found. CMR showed no signs of active or previous myocarditis, decreased left ventricular (EF 13%) and right ventricular (EF 16%) contractility, and myocardial fibrosis. In addition, imaging examinations revealed thrombus in the apical part of the left ventricle and treatment with low-molecular-weight heparin was introduced at a therapeutic dose, which led to regression of the lesion. Due to lack of thrombus in follow-up examination, defibrillation testing (DFT) was performed. Ten months after S-ICD implantation, the patient died of circulatory failure in the course of COVID-19 pneumonia. At the parents’ request, the device was not collected after the boy’s death. Reading was not possible due to epidemiological reasons.5.A 17-year-old female patient with hypertrophic cardiomyopathy and pre-excitation in ECG (after two-fold RF ablation—radiofrequency catheter ablation), treated with Sotalol, in whom genetic testing revealed heterozygous deletion in the LAMP2 region typical for Danon disease. The findings of the echocardiographic examination were as follows: features of hypertrophic cardiomyopathy, LV diastolic dysfunction, borderline size of LA, and normal LV contractility. CMR showed the presence of a parietal LV thrombus, LV myocardial fibrosis, mixed hypertrophic cardiomyopathy, and LV myocardial noncompaction with normal LV contractility. ECG monitoring reported episodes of ventricular tachycardia and multiple narrow QRS complex tachycardias. Due to the presence of thrombus, no defibrillation test was performed.6.A 16-year-old male patient with hypertrophic cardiomyopathy, whose echocardiographic examination showed extreme concentric left ventricular hypertrophy and almost complete obliteration of the ventricular lumen in systole. In CMR, visible signs of edema within LV myocardium and diffuse areas of myocardial fibrosis/necrosis of non-ischemic etiology were revealed. Self-limiting ventricular tachycardia and prolonged QT with a mean QTc 460 ms were recorded in Holter monitoring.7.A 17-year-old male patient with complex ventricular arrhythmia and arrhythmogenic right ventricular cardiomyopathy. Holter monitoring revealed PVCs with increasing tendency (including single VEBs, 10% pairs of ventricular extrasystoles, and VT episodes). Positive family history—the boy’s brother died suddenly at age 13 years and the boy’s father died at age 40 years. In echocardiography, RV within the upper limit of normal, convex RV free wall, thick septo-marginal trabecula and increased trabeculation of the apical part of the right ventricle, and preserved systolic function of the right ventricle were found. The main criteria for the diagnosis of arrhythmogenic right ventricular cardiomyopathy (segmental wall dyskinesis, enlarged RV > 110 mL/m^2^, EF < 40%) in CMR were fulfilled.

S-ICD was implanted for secondary prevention in four patients:8.A 14-year-old female patient, successfully defibrillated with an Automated External Defibrillator (AED) after outpatient cardiac arrest. The incident took place in a mall while shopping. Ventricular fibrillation was documented in AED records. Imaging examinations (ECHO) showed normal morphology of the heart structures. Diagnostic tests toward channelopathy were performed: no signs of prolonged QT were found in a series of ECG tests. Exercise stress test was negative—there was no arrythmia during the test and QTc in the 4th minute of rest was normal. Subsequently, a drug provocation test was conducted with adrenaline and ajmaline—both were negative. Therefore, idiopathic ventricular fibrillation was diagnosed. Beta-blocker treatment was introduced with good drug tolerance. The patient was very slim and implantation was technically difficult, but the cosmetic effect was very good (Figure 3 and Figure 4).9.A 12-year-old female patient with long QT syndrome (5 points on the Schwartz scale), confirmed by genetic testing, treated with propranolol. While playing in a pool (despite the ban, she bathed and played ball in the pool), the girl lost consciousness for about 2 min. The loss of consciousness was sudden and was not preceded by palpitations or a feeling of weakness, visual disturbances, or dizziness. During the period of unconsciousness, the girl first presented with tautness; she was pale, limp, and did not respond to stimuli. After being taken out of the pool, she was still unconscious, without a pulse. After consciousness returned, there was no confusion or excessive sleepiness. Since then, QTc 0.46–0.5 s has been observed. The girl’s older brother is also a patient of the Clinic; he was diagnosed with LQTS, which was genetically confirmed. During the course of diagnosis of both children, their mother was also diagnosed with prolongation of QT interval, as well as their grandmother. During the girl’s exercise stress test, QTc at the 4th minute after exercise was 0.49 s. Due to her very slim body structure and low weight (38 kg), the procedure was technically difficult, but it was possible to create an intramuscular pocket. The procedure was performed using a two-incision technique. There were no complications. On day 2, the patient was mobilized, and after about a week during a control visit, she presented without pain with almost the entire range of left-hand movement was recovered (Figure 5).10.A 16-year-old male patient with sudden cardiac arrest that occurred at school during a sports lesson—the child collapsed without any warning symptoms after having performed physical activity. The teacher started CPR. At the scene of the incident, paramedics diagnosed the boy with ventricular fibrillation and performed defibrillation three times, restoring the sinus rhythm. Family history of SCD was negative. ECG showed normal sinus rhythm, with normal QTc and early repolarization features in leads II, III, aVF, V5, and V6. Holter monitoring ECG showed no arrhythmia. ECHO showed normal LV size and contractility. An exercise treadmill test was performed—no arrhythmia was observed with normal QTc at 4 min of rest. ECG with high right ventricular leads and provocative test with flecainide were performed and were both negative. Angio CT of the coronary arteries revealed a muscular bridge over the course of the left anterior descending coronary artery without narrowing in coronarography with nitroglycerines, with no ischemic features and no myocardial edema in CMR.11.A 16-year-old male patient diagnosed with Brugada syndrome, implanted with an ICD system, qualified for S-ICD implantation due to battery depletion and increasing defibrillation lead resistance. He had a history of several episodes of unconsciousness up to the age of 3 years and, at age 9 years, during an episode of SCA, the boy lost consciousness on the beach; he was assisted by a cardiologist present there, who started CPR. During hospitalization after the episode, he was diagnosed with Brugada syndrome. He was implanted with an ICD and one year after implantation, there was an episode of adequate defibrillation. It was decided that the transvenous system would be temporarily left and an S-ICD device implanted and, after healing, a decision would be made on the intracardiac lead removal.

There were no early or late complications related to the procedure in any of the patients (Table 1 and Table 2).

## 4. Discussion

S-ICD constitutes a new treatment perspective for patients with primary electrical or structural heart disease, preventing them from SCD, while the heart and vessels remain unaffected. Its advantages to young patients are aesthetically pleasing results. The S-ICD system diminishes the risk of the lead insulation damage because of its multistrand cable core lead design and no lumen and no systolic and diastolic cyclic friction within the heart. Implantation of the S-ICD using anatomical landmarks reduces the X-ray dose significantly. The risk of lead-related infective endocarditis, cardiac perforation, and pneumothorax is eliminated. Advanced experimental trials are being conducted to develop a smaller and lower-energy pulse generator system with a miniaturized lead. It is estimated that 35 J of energy should be sufficient for this purpose in small children of 5–10 kg weight [11]. The maximal device energy output is 80 J, which is higher than necessary for defibrillation in low-weight children [11].

It is reported that the overall rates of complications in adults after S-ICD are similar to those of transvenous systems [12,13,14]. There are reductions in lead dysfunction and system infection requiring explanation, observed along with increases in inappropriate shocks. Studies in pediatric patients also raised concerns regarding the generator size and inappropriate shocks [15,16], although more recent data show comparable rates of complications [17,18,19,20].

Use of the S-ICD is indicated in patients who are ineligible for a transvenous system, patients with difficult venous access to the heart due to thrombosis, or congenital anatomical anomalies, after transvenous electrode removal and as a procedure of choice in young patients. It is extremely important to emphasize that, in the event of infection or damage to the S-ICD electrode, its removal and replacement is much easier and safer and with minimal risk of severe complications compared to transvenous electrode removal and replacement. This is an important argument in favor of the S-ICD as the first-choice method in young patients over the TV-ICD [21]. As this is the experience of a pediatric center, the main factors in favor of S-ICD over TV-ICD were young age of the patients and lack of indications for cardiac pacing. Despite growing experience with local anesthesia, due to limited cooperation and anxiety associated with the procedure in children and adolescents, we prefer to perform S-ICD implantation under general anesthesia [5]. In all cases, the pocket for the generator was created between the serratus anterior and the latissimus dorsi muscles. Currently, in our opinion, it is a method with fewer late complications, which gives an excellent cosmetic effect.

There were 11 patients in our group, 8 boys (72.7%) and 3 girls (27.3%), aged from 12 to 17 years, mean 15.5 years. In the monitoring so far, none of our patients have suffered from inadequate defibrillation. To date, we have not observed late complications. The largest study published to date described 115 children and adolescents from 15 centers in the United States. For comparison, the cited study comprised 115 patients with a median follow-up of 32 (19 to 52) months. Their median age was 16.7 years (14.8 to 19.3 years); 29% were female and 71% were male. Fifty-five per cent had a primary prevention indication. The underlying disease substrates were cardiomyopathy (40%), structural heart disease (32%), idiopathic ventricular fibrillation (16%), and channelopathy (13%). The complication rate was 7.8% at 30 days and 14.7% at 360 days. Overall, inappropriate shocks occurred in 15.6% of patients. At implant, 97.9% of patients had successful first-shock conversion with 96% requiring ≤65 J. Appropriate therapy was delivered to 11.2% [17].

A study from another center in Poland was published in 2020, which covered a group of 80 patients with ICDs implanted over a period of 22 years (13). In the study, a group of children, adolescents, and young adults aged 6–21 years with different types of implanted devices was analyzed. S-ICDs were implanted in eight patients, whose average age was 18 years (15–21 years). The exact number of pediatric patients under the age of 18 years with an S-ICD implantation is not clearly stated.

At the time of publication, there are two more patients who have been successfully qualified in our center for the S-ICD procedure: two girls with hypertrophic cardiomyopathy and high risk of SCD. The parents previously refused consent to a venous ICD, so the only option being considered is the S-ICD.

## 5. Conclusions

Our results prove the efficacy of the S-ICD treatment option along with technically simple surgery, which supports its further widespread application in children.

We are convinced that the safety, short term of the procedure, good cosmetic effect, even in slim patients, and the absence of intravascular or intracardiac components offer potential advantages to pediatric patients and those with congenital heart disease.

## Figures and Tables

**Figure 1 ijerph-19-11661-f001:**
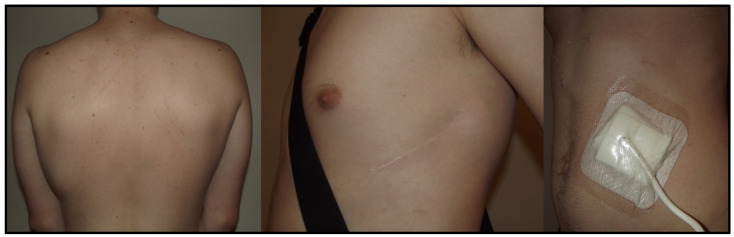
Patient 2. Cosmetic effect after S-ICD implantation in a 17-year-old patient with dilated cardiomyopathy and LVAD HeartMate 3. S-ICD, subcutaneous implantable cardioverter defibrillator; LVAD left ventricular assist device.

**Figure 2 ijerph-19-11661-f002:**
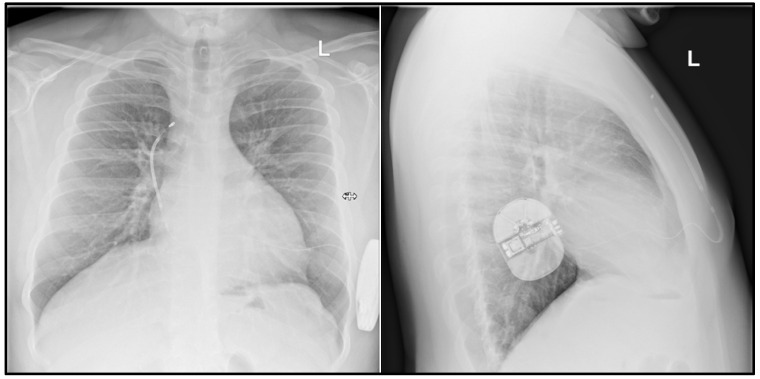
Patient 3. Chest X-ray of an obese patient with electrode placed on the right side of the sternum. This coil positioning is due to the patient’s composition features and obesity as well as the placement of the electrode (coil) on the right side. After implantation, vectors were obtained as during screening and a normal defibrillation test result was obtained.

**Figure 3 ijerph-19-11661-f003:**
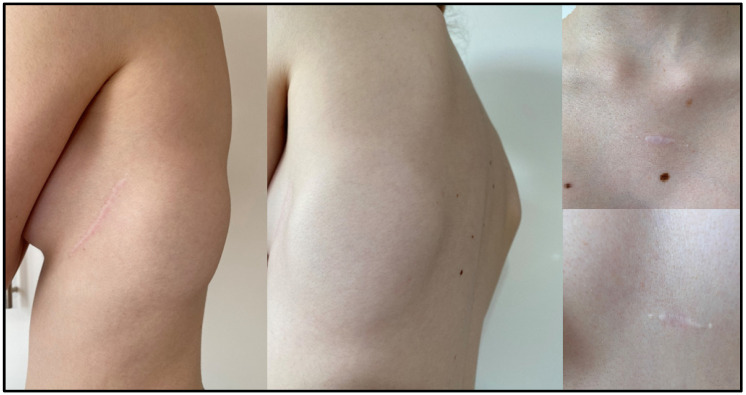
Patient 8. Cosmetic effect after S-ICD implantation in a 14-year-old female patient after cardiac arrest by ventricular fibrillation. S-ICD, subcutaneous implantable cardioverter defibrillator.

**Figure 4 ijerph-19-11661-f004:**
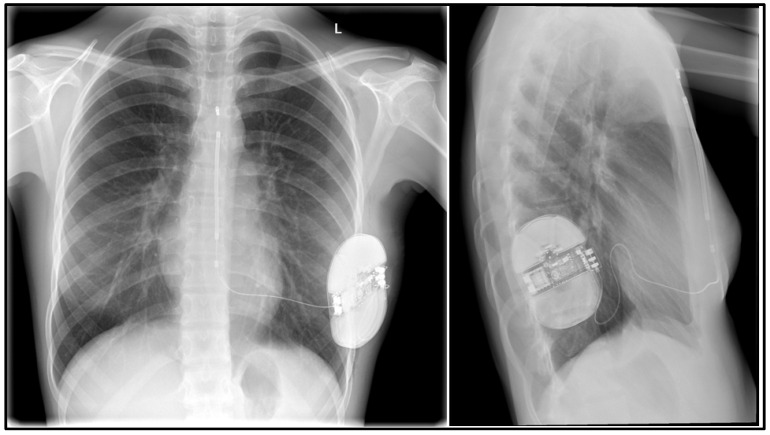
Patient 8. Chest radiograph of a slim female patient.

**Figure 5 ijerph-19-11661-f005:**
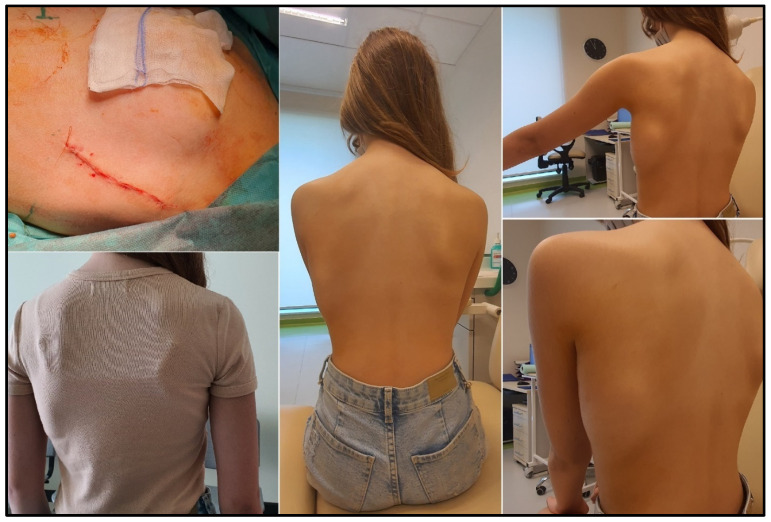
Patient 9. Cosmetic effect after S-ICD implantation in a 12-year-old female patient after cardiac arrest in the course of ventricular fibrillation.

**Table 1 ijerph-19-11661-t001:** Demographics and clinical characteristics of the children’s S-ICD population.

No. of Patient	Sex	Age (Years)	Weight	Height	Indications for S-ICD	SCD Prevention	Prior CIED	NYHA Class	ECHO	CMR	Arrythmia	Indication for Pacing
1.	M	17	65	188	HCM, LVOTO	Primary	No	-	LVEF 82% LVPWd 17–21 mm (n do 10.2). IVSd 22.8 mm (n do 10.7), LVOTO: max. 108–115; mean 55 mmHg	Fibrosis present Max LV: 28 mm IVS: 22 mm	PVCs	No
2.	M	17	50	168	DCM, HF	Primary	No	Chronic HF; NYHA III; LVAD HeartMate3; Patient qualified for HTx	LVEF 24–28%	LVEF = 18%. Scattered areas of LV myocardial fibrosis/necrosis of non-ischemic aetiology	nsVT 130-170/min.	No
3.	M	16	123	177	HCM, LVOTO	Primary	No	-	IVSd 24.3 mm, LVPWd 22.8 mm.; LVOTO 70 mmHg; IAo IIst.	Heterogenous myocardial structure, without areas of fibrosis/necrosis	PVCs	No
4.	M	17	67	166	DCM; HF	Primary	No	Chronic HF; NYHA III; Patient qualified for HTx	LVEF 10–15–25% IMV I/II st. ITV I/II st.	LVEF 13% RVEF 16% Fibrosis of myocardium present	nsVT	No
5.	F	17	76	170	HCM + LVNC Danon disease	Primary	No	-	IVSd 20 mm, LVPWd 16 mm	Mixed cardiomyopathy—HCM and LVNC	nsVT	No
6.	M	16	88	178	HCM	Primary	No	-	IVSd 47.2 mm, LVPWd 29 mm	LV max 35 mm; IVS 48 mm Sings of myocardial edema in LV and scattered areas of fibrosis/necrosis of non-ischemic aetiology	nsVT	No
7.	M	17	73	177	ARVC	Primary	no		right ventricle to the upper limit of normal—RV-EDV in ECHO 3D 194 mL (102 mL/m^2^) vs. LVEDV 174 mL.	large resonance criteria for the diagnosis of ARVC (segmental wall dyskinesia, increased RV > 110 mL/m^2^, EF < 40%).	PVCs, VT	
8.	F	14	47	168	SCA; VF	Secondary	No	-	PMV	no abnormalities	PVC, TdP, VT, IVF	No
9.	F	12	38	152	LQTS	Secondary	No	-	no abnormalities	no abnormalities	Not found	No
10.	M	16	68	173	SCA; VF	Secondary	No	-	no abnormalities	Generalized edema probably related to resuscitation. On follow-up no evidence of edema	Not found	No
11.	M	16	58	172	Brugada syndrome; SCA	Secondary	Yes	-	bicuspid aortic valve	-	VT	No

ARVC, arrhythmogenic right ventricular cardiomyopathy; CIED, cardiac implantable electronic device; CMR, Cardiovascular magnetic resonance imaging; DCM, dilated cardiomyopathy; ECHO, echocardiography; F, female; HCM, hypertrophic cardiomyopathy; HF, heart failure; HTx, heart transplantation; ICD, implantable cardioverter-defibrillator; IMV, mitral valve insufficiency; IVF, idiopathic ventricular fibrillation; IVSd, interventricular septum thickness in diastole; LQTS, long QT syndrome; LVEF, left ventricular ejection fraction; LVNC, left ventricular noncompaction; LVOTO—left ventricular outflow tract obstruction; LVPWd, Left ventricular posterior wall thickness at end-diastole; M, male; nsVT, non-sustained ventricular tachycardia; NYHA, New York Heart Association; PMV, mitral valve prolapse; PVCs, premature ventricular contractions; SCA, sudden cardiac arrest; SCD, sudden cardiac death; S-ICD, subcutaneous implantable cardioverter-defibrillator; TdP, torsades des pointes; SR, sinus rhythm; VF, ventricular fibrillation; VT, ventricular tachycardia.

**Table 2 ijerph-19-11661-t002:** S-ICD implantation—surgery details.

No. of Patient	Anaesthesia	Physicians *	X-rays **	S-ICD Pocket	Incisions	S-ICD Test	Time ***	Operative Time	Complication
1.	General	CS, EP, CS *	Yes	Intermucsular	3	Yes 1st Effective (65 J)	150	60	None
2.	General	CS, EP, PC *	Yes	Intermucsular	3	Yes 1st Effective (65 J)	120	50	None
3.	General	CS, PC, EP *	Yes	Intermucsular	3	2nd Effective (65 J rev.)	120	45	None
4.	General	CS, PC, EP *	Yes	Intermucsular	3	Yes 1st Effective (65 J)	90	40	None
5.	General	CS, PC, EP *	Yes	Intermucsular	3	No	120	50	None
6.	General	CS, PC, EP *	Yes	Intermucsular	3	Yes 1st Effective (65 J)	60	30	None
7.	General	CS, PC, EP *	No	Intermucsular	3	Yes 1st Effective (65 J)	100	30	None
8.	General	CS, PC, EP *	Yes	Intermucsular	3	Yes 1st Effective (65 J)	150	55	None
9.	General	CS, PC, EP *	Yes	Intermucsular	2	Yes 1st Effective (65 J)	90	45	None
10.	General	CS, PC, EP *	Yes	Intermucsular	2	Yes 1st Effective (65 J)	70	30	None
11.	General	CS, PC, EP *	Yes	Intermucsular	3	Yes 1st Effective (65 J)	110	40	None

PC—Pediatric Cardiologist, EP—electrophysiologist, CS—cardiac surgeon, rev.—reversed, * proctoring, ** preprocedural X-ray imaging, *** Time from patient in to patient out (min.) approximated to 10 min.

## Data Availability

The data that support the findings of this study are available from the corresponding author, upon reasonable request.

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
