# Peer review of "Subcutaneous Implantable Cardioverter Defibrillators for the Prevention of Sudden Cardiac Death: Pediatric Single-Center Experience"

_ijerph, 2022, doi:10.3390/ijerph191811661_

Round 1

Reviewer 1 Report

In the manuscript “Subcutaneous Implantable Cardioverter-Defibrillators for the Prevention of Sudden Cardiac Death: Pediatric Single-Center Experience”, Piotr Wieniawski et al. performed a single center-retrospective study analysis of pediatric patients implanted with S-ICD. 

This is an interesting topic and the authors’ opinion resulted helpful in the field, but the article needs to be improved under certain aspects: 

Major considerations: 

1.     In my mind this manuscript should be considered more as a case series than a retrospective analysis. Please, consider this suggestion. 

2.      Table 2. The majority of patients underwent a 3-incision technique implant. Nowadays is much more widespread a 2-incision technique. Why did you prefer this technique? 

3.     In discussion, authors should emphasize some topics:

·      Underline who might receive more benefit from S-ICD respect to transvenous ICD (i.e. patients underwent transvenous lead extraction for CIED infection, patients not eligible for conventional ICD, or as in your report, young patients.). Probably a new section titled “Patients Selection” could be interesting. 

·      Please emphasize the feasibility and safety of S-ICS removal respect to TV-ICD in case of infection/adverse events also citing this article “Neglected lead tip erosion: An unusual case of S-ICD inappropriate shock. J Cardiovasc Electrophysiol. 2020. doi: 10.1111/jce.14746. This argument should be important also in the light of first choice of S-ICD respect to TV-ICD. 

4. English language should be improved, I tried to suggest some changes, but the whole manuscript has to be revised by an English native speaker. Below some examples: 

·      Introduction, line 36: “major problems of modern cardiology”, change in “actually one of major concern in cardiology”

·      Introduction, line 37: “popularization” change in “spread”

·      Introduction, line 40-41: “Up to now the vast majority of high-voltage cardiac implants include a defibrillation lead placed transvenously to the right ventricle” change in “do date, the majority of high-voltage devices are represented by transvenous leads.”

·      Abbreviation for “transvenous ICD” should be “TV-ICD (not T-ICD)”

·      When presenting the patients, do no say “12-year-old girl”. Please use the form “a “12-year-old female patient”

Minor Concerns:

1.     Authors present S-ICD as a now technology “Recently, a new device concept”. Conversely, more than 18 years of clinical data shows that the S-ICD is safe and effective and has comparable performance to TV-ICD. Please mitigate this information.

2.     The patients considered were implanted with S-ICD during 2018 and 2022, the authors should provide at least a statement concerning the safety/efficacy outcome during follow-up i.e. “after a mean follow-up time of XXX…”.

3.     Material and methos, lines 68-77: “all patients…cardiologist” this part is redundant, please summarize or eliminate it.

4.     Figure 2, patient 3: in AP chest X-ray, the coil seems not perfectly straight. Why this? It was due to patient’s characteristics? (obese patient). Did it affected the efficacy? Briefly mention it. 

Author Response

Response to the Reviewer

We would like to express our thanks to the Expert reviewer  for the  helpful comments for correction and modification of our manuscript entitled:

Subcutaneous Implantable Cardioverter-Defibrillators for the Prevention of Sudden Cardiac Death: Pediatric Single-Center Experience”,

In response to yours suggestions (our answers are written in italics):

In the manuscript “Subcutaneous Implantable Cardioverter-Defibrillators for the Prevention of Sudden Cardiac Death: Pediatric Single-Center Experience”, Piotr Wieniawski et al. performed a single center-retrospective study analysis of pediatric patients implanted with S-ICD. 

This is an interesting topic and the authors’ opinion resulted helpful in the field, but the article needs to be improved under certain aspects: 

Major considerations: 

In my mind this manuscript should be considered more as a case series than a retrospective analysis. Please, consider this suggestion. 

We took this into consideration. As the group of patients with implanted S-ICD grew and became more and more diverse therefore they do not fulfil the definition of case series. We present a group of patients dividing them into subgroups with different problems and indication. That justifies the original form of the article.

Table 2. The majority of patients underwent a 3-incision technique implant. Nowadays is much more widespread a 2-incision technique. Why did you prefer this technique? 

We have had kits for two-incision implantation available only for the last five implantations. At the moment, our first choice is the two-incision technique. If the patient is very obese, we prefer the three-incision technique to avoid electrode dislocation. It happened twice, that sheath has been compressed by subcutaneous tissues and the insertion of the electrode has risked damaging it. In such cases, third micro-incision is made solely for the purpose of pulling the electrode through. This is due to the fact that in children and adolescents, soft tissues between skin and the ribs are often very compact and, after removal of the guidewire, the tissues compress the sheath, which flattens and folds over the ribs.

We reported this problem to the company representative.

In discussion, authors should emphasize some topics:

Underline who might receive more benefit from S-ICD respect to transvenous ICD (i.e. patients underwent transvenous lead extraction for CIED infection, patients not eligible for conventional ICD, or as in your report, young patients.). Probably a new section titled “Patients Selection” could be interesting. Please emphasize the feasibility and safety of S-ICS removal respect to TV-ICD in case of infection/adverse events also citing this article “Neglected lead tip erosion: An unusual case of S-ICD inappropriate shock. J Cardiovasc Electrophysiol. 2020. doi: 10.1111/jce.14746”. This argument should be important also in the light of first choice of S-ICD respect to TV-ICD. 

added to the discussion:

Use of the S-ICD is indicated in patients who are ineligible for a transvenous system, patients with difficult venous access to the heart due to thrombosis or congenital anatomical anomalies, after transvenous electrode removal and as a procedure of choice in young patients. It is extremely important to emphasize that, in the event of infection or damage to the S-ICD electrode, its removal and replacement is much easier and safer and with minimal risk of severe complications compared to transvenous electrode removal and replacement. This is an important argument in favor of the S-ICD as the first choice method in young patients over the TV-ICD

As this is the experience of a paediatric centre, the main factors in favor of S-ICD over TV-ICD were young age of the patients and lack of indications for cardiac pacing.

We added relevant literature to the discussion.

Mitacchione G, Schiavone M, Gasperetti A, Viecca M, Curnis A, Forleo G.B. Neglected lead tip erosion: An unusual case of S-ICD inappropriate shock. J. Cardiovasc. Electrophysiol. 2020, 31, 3322–3325

English language should be improved, I tried to suggest some changes, but the whole manuscript has to be revised by an English native speaker. Below some examples: 

The article has been proofread by a native speaker who provides medical translations.

  • Introduction, line 36: “major problems of modern cardiology”, change in “actually one of major concern in cardiology”

      Modified as suggested

  • Introduction, line 37: “popularization” change in “spread”

      Modified as suggested

  • Introduction, line 40-41: “Up to now the vast majority of high-voltage cardiac implants include a defibrillation lead placed transvenously to the right ventricle” change in “do date, the majority of high-voltage devices are represented by transvenous leads.”

Modified as suggested

Abbreviation for “transvenous ICD” should be “TV-ICD (not T-ICD)”

Modified as suggested

  • When presenting the patients, do no say “12-year-old girl”. Please use the form “a “12-year-old female patient”

revised as suggested in all places in the text

Minor Concerns:

Authors present S-ICD as a now technology “Recently, a new device concept”.

Conversely, more than 18 years of clinical data shows that the S-ICD is safe and effective and has comparable performance to TV-ICD. Please mitigate this information.

The sentence fragment has been removed. As rightly noted it misleads the reader.

The patients considered were implanted with S-ICD during 2018 and 2022, the authors should provide at least a statement concerning the safety/efficacy outcome during follow-up i.e. “after a mean follow-up time of XXX…”.

We have not observed early nor late complications so far, no patient has required reoperation or device replacement, and we have not observed inadequate nor appropriate interventions. One patient died of heart failure, which we wrote about in the paper.

Material and methos, lines 68-77: “all patients…cardiologist” this part is redundant, please summarize or eliminate it.

As suggested, it was shortened and replaced with:

All patients were screened in accordance with currently applicable procedures and standards.

Figure 2, patient 3: in AP chest X-ray, the coil seems not perfectly straight. Why this? It was due to patient’s characteristics? (obese patient). Did it affected the efficacy? Briefly mention it. 

added in figure description:

This positioning is due to the patient's composition features and obesity as well as the placement of the electrode (coil) on the right side. After implantation, vectors were obtained as during screening and a normal defibrillation test result was obtained.

Sincerely, authors

Reviewer 2 Report

The S-ICD is a relatively new method of preventing sudden cardiac death. The listed advantages are certainly present. However, in this work, the conclusion is that S-ICD implantation is a successful method in the prevention of sudden cardiac death. Colleagues performed the procedure and implantation beautifully and had no complications of the procedure. But to conclude whether the method is successful, it would be good to know a specific follow-up. Also, the S-ICD does not have the option to treat malignant arrhythmias with ATP. It would be good to explain why (besides aesthetics and reducing the possibility of intravascular complications) it was decided to install the S-ICD.

Author Response

Response to the Reviewer

We would like to express our thanks to the Expert reviewer  for the  helpful comments for correction and modification of our manuscript entitled:

Subcutaneous Implantable Cardioverter-Defibrillators for the Prevention of Sudden Cardiac Death: Pediatric Single-Center Experience”,

In response to yours suggestions (our answers are written in italics ):

The S-ICD is a relatively new method of preventing sudden cardiac death. The listed advantages are certainly present. However, in this work, the conclusion is that S-ICD implantation is a successful method in the prevention of sudden cardiac death. Colleagues performed the procedure and implantation beautifully and had no complications of the procedure. But to conclude whether the method is successful, it would be good to know a specific follow-up. Also, the S-ICD does not have the option to treat malignant arrhythmias with ATP. It would be good to explain why (besides aesthetics and reducing the possibility of intravascular complications) it was decided to install the S-ICD.

We have not observed early nor late complications so far, no patient has required reoperation or device replacement, and we have not observed inadequate nor appropriate interventions. One patient died of heart failure, which we wrote about in the paper.

added to the discussion:

Use of the S-ICD is indicated in patients who are ineligible for a transvenous system, patients with difficult venous access to the heart due to thrombosis or congenital anatomical anomalies, after transvenous electrode removal and as a procedure of choice in young patients. It is extremely important to emphasize that, in the event of infection or damage to the S-ICD electrode, its removal and replacement is much easier and safer and with minimal risk of severe complications compared to transvenous electrode removal and replacement. This is an important argument in favor of the S-ICD as the first choice method in young patients over the TV-ICD

As this is the experience of a paediatric centre, the main factors in favor of S-ICD over TV-ICD were young age of the patients and lack of indications for cardiac pacing.

Sincerely, authors

Round 2

Reviewer 1 Report

I congratulate with the authors for the effort that improved the overall quality of the manuscript. No further comments.